# The impact of community based interventions for the prevention and control of soil-transmitted helminths: A systematic review and meta-analysis

**Sommy C. Ugwu**[1]*, **Michael O. Muoka**[2], **Clara MacLeod**[1], **Sarah Bick**[1], **Oliver Cumming**[1], **Laura Braun**[1]

**1** London School of Hygiene and Tropical Medicine, London, United Kingdom, **2** Zenith Medical and Kidney Centre, Abuja, Nigeria

* sommy.c.ugwu@gmail.com

**Data Availability Statement:** All data reported in the manuscript are provided as part of the submitted article or in the supplementary

## Abstract

Soil-transmitted helminths (STHs) are among the most common human infections worldwide and a major cause of morbidity. They are caused by different species of parasitic worms and transmitted by eggs released in faeces or when hookworm larvae penetrate the skin. The main control strategy in endemic regions is periodic treatment with deworming medication. In the last 10 years, there has been a scale-up of prevention and control activities with a focus on community-based interventions (CBIs). This review aims to systematically analyse the impact of CBIs on the prevalence and infection intensity of STHs. A systematic review was published on this topic in 2014, but there have subsequently been several new studies published which are included in this review. Electronic database search of MEDLINE (Ovid), Global Health Online (Ovid), Cochrane Library, Embase (Ovid) and Web of Science was conducted. Titles, abstracts, and full texts were screened by two independent reviewers according to predefined eligibility criteria. Data were extracted and a meta-analysis of included studies was conducted. A total of 11,954 de-duplicated titles were screened, and 33 studies were included in the review. 14 focussed on community-wide treatment, 11 studies investigated school-based interventions, and 3 studies investigating both. Results suggest that CBIs are effective in reducing the prevalence of Hookworm, *Trichuris trichiura* and *Ascaris lumbricoides*. School-based treatment and community-wide treatment, as well as annual and semi-annual deworming, all reduce STH prevalence significantly. Mass drug administration was effective in reducing the infection intensity of Hookworm (Mean difference: -211.36 [95% CI: -519.12, 96.39]), *Trichuris trichiura* (-736.69 [-1349.97, -123.42]) and *Ascaris lumbricoides* (-2723.34 [-5014.85, -431.84]). The results suggest that CBIs are effective in reducing the prevalence and intensity of STH infections. While most studies delivered preventive chemotherapy (PC), few studies explored the impact of interventions such as water, sanitation, and hygiene (WASH) or health education, which may be essential in preventing reinfection after PC.

documentation. The supplementary file submission contains all raw data required to replicate the results of this study.

**Funding:** The authors received no specific funding for this work.

**Competing interests:** The authors have declared that no competing interests exist.

## Introduction

Soil-transmitted helminths (STHs) are a sub-group of neglected tropical diseases (NTDs) that continue to be transmitted in Africa, the Americas, and East Asia [1]. STHs include round-worms (*Ascaris lumbricoides*), whipworms (*Trichuris trichiura*) and hookworms (*Ancylostoma duodenale* and *Necator americanus*) [2]. They represent a vast public health problem, with approximately 1.5 billion people infected worldwide [3]. Over 70% of the mortality burden lies in sub-Saharan Africa [3]. Infection occurs through ingestion of embryonated eggs or when larvae in the soil penetrate the skin [4].

Children, pregnant women, and young adults living in low- and middle-income countries are disproportionately affected by STHs and experience more severe sequelae [5]. The world-wide burden of STH infection was estimated to be responsible for up to 39 million disability-adjusted life years (DALYs), resulting from diarrhoea, anaemia, malnutrition, physical and mental growth retardation and other adverse health outcomes [1,6]. These infections are per-petuated by, and reinforce social determinants of poverty, illiteracy, poor nutrition, and a lack of access to health education as well as safe water, sanitation and hygiene (WASH) [7].

The World Health Organization's (WHO) 2021–2030 road map for NTDs targets the elimi-nation of STHs as a public health problem by 2030 (defined as <2% of moderate-to-heavy intensity infection) [7]. Regular preventive chemotherapy (PC) with the anthelminthic drugs albendazole or mebendazole is recommended by the WHO in areas where prevalence exceeds 20%. This can control morbidity, but reinfection is likely to continue if populations continue to live in highly contaminated environments due to inadequate WASH infrastructure. Inter-ventions including health education and WASH are therefore also recommended [7]. Progress in delivering PC has been in part driven by the donations of billions of anthelmintic drugs from the pharmaceutical industry [8], which contribute to the cost-effectiveness of control programs. Drugs are often delivered through schools or the community, which provides an effective platform for drug administration.

A community-based intervention (CBI) includes any intervention delivered in a commu-nity setting, defined here as domestic, public and institutional settings, rather than traditional hospital-based care delivered in health facilities. In reference to STHs, this can include inter-ventions such as health education, WASH, distribution of shoes and PC. Delivery mode of PC includes community-wide treatment (CWT) and school-based treatment (SBT). The advan-tage of SBT is that it makes use of existing infrastructure to target school-aged children, the age group most at risk of STH infection. Nonetheless, CWT may be important for interrupting the transmission of STHs since infected adults or pre-school aged children not reached through SBT remain infected. WHO guidelines recommend annual PC for school-aged chil-dren in areas where STH prevalence in children is between 20%-50% and semi-annual PC in communities where prevalence exceeds 50% [9].

CBIs can be integrated or non-integrated. Integrated CBIs are delivered as part of a pre-existing routine prevention and control program. Interventions can be integrated with WASH or drug treatment targeting other diseases. Integrated WASH interventions refer to those that augment PC with WASH while integrated chemotherapy for NTDs refers to a combination of separate PC programs targeting multiple NTDs including schistosomiasis, lymphatic filariasis, onchocerciasis, and trachoma.

A previous systematic review conducted in 2014 found that CBIs were effective in reducing the prevalence and infection intensity of STHs [10]. The review also found that CBIs reduced the prevalence of anaemia but had no impact on anthropometric indices. There is a need to analyse recent data, especially considering the scaling up of prevention and control programs in the last decade as well as evidence of reduced drug efficacy in an area with high drug pressure [11].

## Aims

The aim of this review is to critically evaluate the effectiveness of CBIs for the prevention and control of STHs. The previous review on the same topic was published in 2014 with literature search conducted in 2013 [10].This review aims to synthesise the latest evidence, shedding light on the evolving landscape of community-based interventions.

## Methods

### Search strategy

Search terms relating to the concepts of this review, namely "Community based interventions" and "Helminth disease", were included. The search terms were refined following a test search. The full list of search terms with synonyms is provided in S1 Data. Electronic database searches were conducted in MEDLINE (Ovid), Global Health Online (Ovid), Cochrane Library, Embase (Ovid) and Web of Science. Search terms were mapped to MeSH terms in MEDLINE, Cochrane and Embase libraries. The search field was limited to title and abstract. Searches were conducted on 30th November, 2023 and were restricted to articles published from 2013, which is when the search of the previously published review was conducted [10]. Additionally, reference lists of included studies were screened manually to identify relevant studies. The review protocol was not registered.

### Eligibility criteria

Peer-reviewed articles published after April 2013 were eligible for inclusion if they reported on the effectiveness of the community-based delivery of interventions for STHs. Randomised, quasi-randomised and before-and-after studies were included in this review, in which intervention was carried out in the community and reported outcomes (prevalence or intensity of infection) could be compared to baseline measures or a control group, and reported per species, not combined. The meta-analysis included only randomised trials. Non-peer reviewed articles or grey literature were excluded. Interventions that were delivered in the hospital setting were not eligible for inclusion since they are not considered CBIs. Cross-sectional studies, case reports, conference abstracts, qualitative studies and secondary research articles were also excluded. Studies were not excluded based on language of publication.

### Study screening and selection

Following database search, all results were exported to Mendeley reference manager software and de-duplicated. Two reviewers (SU, CM) independently screened titles and abstracts for eligibility. The full texts of remaining studies were sought and screened by the two reviewers. Any unsolved discrepancies were discussed with the third reviewer (LB). Reasons for exclusion of studies were documented at each step.

### Quality appraisal of included studies

An adapted version of the Cochrane risk of bias tool (RoB2) [12] was used for quality appraisal of randomised trials, while the risk of bias in non-randomized studies—of Interventions tool (ROBINS-I) [13] was adapted and used to appraise non-randomised studies. The included studies were categorised as having very poor, poor, fair or good quality. Two co-authors (SU and MM) independently assessed the quality, and any discrepancies were resolved with the third author (LB).

### Outcome measures

The primary outcomes are STH prevalence (reported per species) and intensity of infection (eggs per gram of stool). Secondary outcomes are nutritional outcomes such as anaemia, stunting and any reported nutritional outcomes and growth indices.

### Data extraction

A data extraction table was created in Excel based on the data collection form available from The Cochrane Collaboration [14]. Study outcome data and characteristics, including study aims, design, methods, setting, sample size, participant characteristics, time frame, type of intervention, limitations and funding sources, were systematically extracted for each study.

### Data synthesis and analysis

Following data extraction, a meta-analysis was conducted using comparable outcome data from included studies in Review Manager version 5.4 software [15]. Only studies reporting primary outcomes infection for at least one STH species are included in the meta-analysis. For randomised trials, endline intervention and control outcome data are compared. For uncontrolled studies, baseline and endline are compared. Where multiple follow-ups were conducted, the data from the last follow-up was extracted.

Effect measures were reported as relative risk (RR) for prevalence and standard mean difference (SMD) for infection intensity with 95% confidence intervals (CI). The arithmetic mean of eggs per gram of stool (epg) was used as a measure for intensity of infection. This is calculated using data from all participants, unlike the geometric mean which only includes data from positive egg counts (>0 epg). If not provided, standard deviations for infection intensity were calculated using the sample size and confidence intervals provided, as per the Cochrane Handbook [14].

Sub-group analyses were conducted on treatment frequency (annual vs semi-annual) as well treatment delivery (school-based vs community-based). Analyses were run in Review Manager, pooling and comparing the relative risk of sub-groups. For studies that could not be included in the meta-analysis, a narrative synthesis is conducted.

## Results

The electronic search of five databases resulted in a total of 25,546 records. No additional studies were found from citation follow-up and search of reference list of included studies. All records were imported into Mendeley reference manager where 13,592 duplicates were removed. The titles and abstracts of the remaining articles were screened according to eligibility criteria including full text review of 129 articles. In total, 33 articles met the inclusion criteria and were included in this synthesis. The PRISMA flowchart presents further details (Fig 1).

### Characteristics of included studies

33 studies are included in this review. Most of the included studies focused on PC delivering mebendazole or albendazole as the primary CBI (25/33). Eleven studies investigated school-based interventions while 14 focussed on community-based drug treatments. Only Four studies investigated WASH, and one investigated shoe wearing. The majority of included studies were conducted in Kenya (18%), followed by the Democratic Republic of Congo (12%), Malaysia (9%) and Timor-Leste (9%). All studies reported on hookworm, whereas two studies did not report on *A. lumbricoides* or *T. trichiura*. Most studies (20/33) used a before-and-after study design and Six were cluster randomised controlled trials (cRCT). Key characteristics of

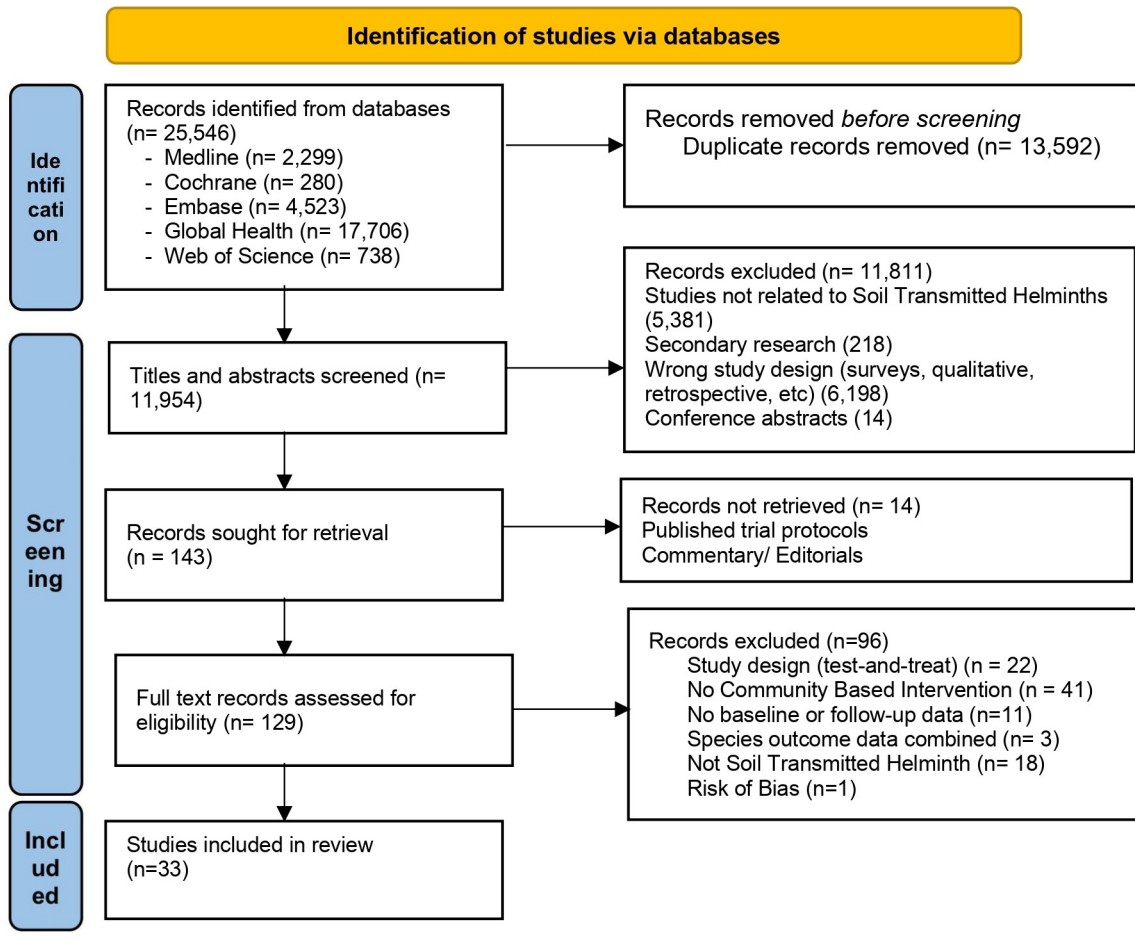

**Fig 1. PRISMA diagram.**

included studies are presented in Table 1. The stool analysis methods used in included studies are outlined in Table 2 below.

## Quality assessment of included studies

Seven studies were judged to have good quality, fourteen as fair, eleven as poor and one as very poor. The study with very poor quality used convenience sampling to select participants and had issues with respecting the maximum slide reading time. The completed risk of bias table is available in S1 Data.

## Effect of PC on STH prevalence and intensity

Five trials are included in the meta-analysis. Pooled analyses show that following PC, the prevalence is reduced by 6% for hookworm (RR: 0.94 95% CI: [0.64, 1.38]), 13% for *A. lumbricoides* (RR: 0.87 95% CI: [0.34, 2.24]) and 23% for *T trichiura* (RR: 0.77 95% CI: [0.63, 0.93]) (Fig 2). The total pooled effect estimate is 0.85 [95% CI: 0.67, 1.09] for all STHs combined. The forest plot (Fig 2) indicates that for *T trichiura*, the pooled estimate of effect demonstrates a significant impact on prevalence, as the CI does not cross zero. Conversely, for *A. lumbricoides* and Hookworm, the CI of the pooled estimates cross zero, suggesting no significant effect on

**Table 1. Characteristics of included studies.**

| Author | Country | Study Setting | Study Design | Sample size, Age range | Species and outcome | Type of Intervention | Quality |
|---|---|---|---|---|---|---|---|
| Freeman et al. 2013 [16] | Kenya | Nyanza Province, western Kenya | Cluster Randomised Controlled Trial | 913, 7–13 years old | Hookworm *Ascaris* spp. *Trichuris trichiura,* | School-based Water, Sanitation and Hygiene | Poor |
| Gyorkos et al. 2013 [17] | Peru | Belen | Cluster Randomised Controlled Trial | 1,089, school aged children | Hookworm *Ascaris* spp. *Trichuris trichiura,* | School-based Health Education package | Good |
| Al-Delaimy et al. 2014 [18] | Malaysia | Lipis district, Pahang, | open-label controlled intervention trial | 317 children, 6–12 yrs | Hookworm *Ascaris* spp. *Trichuris trichiura,* | School-based Health Education package | Fair |
| Al-Mekhlafi et al. 2014 [19] | Malaysia | Pos Betau, Pahang, | Randomised Controlled Trial | 250 chhildren; 7–12 yrs | Hookworm *Ascaris* spp. *Trichuris trichiura,* | School-based integrated High dose vitamin A and Deworming | Good |
| Njenga et al. 2014 [20] | Kenya | Matuga District, Kwale County | Before-after study | 1022, 7–12 years old | Hookworm | School-based deworming | Poor |
| Nikolay et al. 2015 [21] | Kenya | schools surveyed in western Kenya | Before-after study | 21,528, school-aged children | Hookworm *Ascaris* spp. *Trichuris trichiura,* | School-based deworming | Good |
| Pion et al. 2015 [22] | Democratic republic of Congo | Seke Pembe | Before-after study | 773 ≥ 5 years old | Hookworm *Ascaris* spp. *Trichuris trichiura,* | Community Wide Treatment | Poor |
| Sunish et al. 2015 [23] | India | Villupuram district of Tamil Nadu State | before-after study | 646 9–10 years old | Hookworm *Ascaris* spp. *Trichuris trichiura,* | School-based deworming | Poor |
| Okoyo et al. 2016 [24] | Kenya | Western region, Rift Valley, Coast and Nyanza. | Before-after study | 21,432, 2–9 years old | Hookworm *Ascaris* spp. *Trichuris trichiura,* | School-based deworming | Poor |
| Ash et al. 2017 [25] | Vietnam | province of Phongsaly; | Non-randomised trial | 375, >6 years old | Hookworm *Ascaris* spp. *Trichuris trichiura,* | Community Wide Treatment | Fair |
| Echazu et al. 2017 [26] | Argentina | Tartagal, Salta province | Community-based pragmatic non-randomised trial | 2,685, No age limitation | Hookworm *Ascaris* spp. *Trichuris trichiura,* Anaemia Nutrition status | Community Wide Treatment | Fair |
| Paige et al. 2017 [27] | Uganda | Kabarole District, western Uganda | Controlled before-after study | 245, >4 years old | Hookworm | Provision of footwear with public health iconography | Poor |

*(Continued)*

**Table 1.** (Continued)

| Author | Country | Study Setting | Study Design | Sample size, Age range | Species and outcome | Type of Intervention | Quality |
|---|---|---|---|---|---|---|---|
| Pion et al. 2017 [28] | Democratic republic of Congo | Seke Pembe | Before-after study | 462, ≥5 years or older | Hookworm *Ascaris* spp. *Trichuris trichiura*, | Community Wide Treatment | Fair |
| Clarke et al. 2018 [29] | Timor-Leste | Aileu and Manufahi Municipalities | Non-randomised cluster intervention trial | 522, School-aged | Hookworm *Ascaris* spp. *Trichuris trichiura*, Anaemia Nutrition status | School-, and community-based integrated Water, Sanitation and Hygiene, and Deworming | Fair |
| Bronzan et al. 2018 [30] | Togo | Binah district | Before-after | 17,100, children no age range | Hookworm *Ascaris* spp. *Trichuris trichiura*, | Integrated Community Wide Treatment | Very poor |
| Hürlimann et al. 2018 [31] | Côte d'Ivoire | South-central Côte d'Ivoire | Controlled before-after study | 1894, no age range | Hookworm *Ascaris* spp. *Trichuris trichiura*, | Community Wide Treatment + Sanitation and health education | Fair |
| Dunn et al. 2019 [32] | Myanmar | Yangon and Ayeyarwaddy region | Longitudinal follow-up study | 523, no age restriction | Hookworm *Ascaris* spp. *Trichuris trichiura*, | Community Wide Treatment | Fair |
| Lemos et al. 2019 [33] | Angola | Cabungo | Before-after study | 209, 2–15 years old | Hookworm *Ascaris* spp. *Trichuris trichiura*, Anaemia | Integrated Water, Sanitation and Hygiene, and Community Wide Treatment | Poor |
| Mwandawiro et al. 2019 [34] | Kenya | Western region, Rift Valley, Coast and Nyanza. | Before-after study | 21,528, 2–24 years old | Hookworm *Ascaris* spp. *Trichuris trichiura*, | School-based deworming | Fair |
| Vas Nery et al. 2019 [35] | Timor-Leste | Manufahi municipality | Cluster Randomised Controlled Trial | 1,947; no age limitation | Hookworm *Ascaris* spp. *Trichuris trichiura*, | Integrated Water, Sanitation and Hygiene, and Deworming | Fair |
| Pullan et al. 2019 [36] | Kenya | Kwale County | Cluster Randomised Controlled Trial | 19,684, ≥ 2 years old | Hookworm *Ascaris* spp. *Trichuris trichiura*, | School-based and Community Wide Treatment | Good |
| Loukouri et al. 2020 [37] | Côte d'Ivoire | Forest zone of eastern Côte d'Ivoire | Before-after study | 2,022, no age range | Hookworm | Community Wide Treatment | Fair |
| Chen et al. 2021 [38] | China | Zhaoyuan, Yangshan, Tongcheng, Guixi, Yueyang, Rongxian, Tunchang, Danling, Kaiyang, and Xiangyun provinces | Before-after study | 20,408, no age range | Hookworm *Ascaris* spp. *Trichuris trichiura*, | Integrated Mass Drug Administration, Health Education and Sanitation | Fair |

(*Continued*)

**Table 1.** (Continued)

| Author | Country | Study Setting | Study Design | Sample size, Age range | Species and outcome | Type of Intervention | Quality |
|--------|---------|---------------|--------------|------------------------|---------------------|----------------------|---------|
| Pion et al. 2021 [39] | Democratic republic of Congo | Mbunkimi and Misay | Before-after study | 1,266, ≥ 5 years old | Hookworm *Ascaris* spp. *Trichuris trichiura*, | Community Wide Treatment | Fair |
| Eneanya et al. 2021 [40] | Liberia | Maryland County, southeastern Liberia | Before-after study | 7,838, ≥ 5 years old | Hookworm *Ascaris* spp. *Trichuris trichiura*, | Community Wide Treatment | Poor |
| Eneanya et al 2022 [41] | Liberia | Lofa County, northern Liberia | Before-after study | 3464, ≥ 5 years old | Hookworm *Ascaris* spp. *Trichuris trichiura*, | Community Wide Treatment | Fair |
| Gebrezgabiher et al. 2022 [42] | Ethiopia | Yeki district, SNNPR, southwest Ethiopia | Before-after | 308, ≥10 years old | Hookworm *Ascaris* spp. *Trichuris trichiura*, | Community Wide Treatment | Poor |
| Landeryou et al. 2022 [43] | Ethiopia | Bolossa Sore | Longitudinal follow-up study | 600, no age restriction | Hookworm *Ascaris* spp. *Trichuris trichiura*, | Integrated Community Wide Treatment, Water, Sanitation and Hygiene, and Behaviour change communication | Fair |
| Muslim & Lim, 2022 [44] | Malaysia | Peninsular | Longitudinal follow-up study | 416, no age range | Hookworm *Ascaris* spp. *Trichuris trichiura*, | Community Wide Treatment | Poor |
| Pion et al. 2022 [45] | Democratic republic of Congo | Seke Pembe | Before-after study | 383, 11–14 years old | Hookworm *Ascaris* spp. *Trichuris trichiura*, | Community Wide Treatment | Poor |
| Dyer et al. 2023 [46] | Vietnam | Dak Lak Province | Cluster Randomised Controlled Trial | 4,955 school arm, 5,093 community arm, | Hookworm | School-based and Community Wide Treatment | Good |
| Le et al. 2023a [47] | Solomon Islands | Western Province | Cluster Randomised Controlled Trial | 2,009, no age range | Hookworm *Ascaris* spp. *Trichuris trichiura*, | Community Wide Treatment | Good |
| Le et al. 2023b [48] | Timor-Leste | Dili, Ermera, and Manufahi municipalities | Before-after study | 541, school-aged children | Hookworm *Ascaris* spp. *Trichuris trichiura*, | School-based deworming | Good |

prevalence. Although one study had a wide effect margin, most found that PC reduced the prevalence of STHs. Two trials reported a significant increase in prevalence for hookworm [19,35].

Some results from the before-after studies deviate from expected reduction in STH prevalence. Two studies from Liberia reported a statistically significant increase in *A. Lumbricoides*

**Table 2. Stool analysis method used in included studies.**

| Study Identification | Time to read | Analysis technique | Sample processed per participant | Quality control method |
|---|---|---|---|---|
| Freeman et al. 2013 [16] | Within 1 hour | Kato-Katz | Double | Performed on 10% of slides |
| Gyorkos et al. 2013 [17] | Within 24 hours | KatoKatz | Single | Performed on 25% of all slides. |
| Al-Delaimy et al. 2014 [18] | Within 5 hours | Kato-Katz; Harada Mori culture technique | Not reported | Not reported |
| Al-Mekhlafi et al. 2014 [19] | Not reported | Kato-Katz and Harada Mori techniques | Not reported | Not reported |
| Njenga et al. 2014 [20] | Within 30–45 minutes | Kato-Katz | Double | 10% were re-read |
| Nikolay et al. 2015 [21] | Not reported | Kato-Katz | Double | Not reported |
| Pion et al. 2015 [22] | Within 24 hours | modified Kato–Katz | Double | Not reported |
| Sunish et al. 2015 [23] | Not reported | Kato-Katz | Not reported | Not reported |
| Okoyo et al. 2016 [24] | Not reported | Kato-Katz | Double | 10% were re-read |
| Ash et al. 2017 [25] | Not reported. | McMaster method | Not reported | Not reported |
| Echazu et al. 2017 [26] | Within 24 hours | Harada-Mori filter-paper culture; Baermann concentration and McMaster egg counting method | Not reported | Not reported |
| Paige et al. 2017 [27] | within 24 hours | Formalin-ethyl acetate sedimentation | Not reported | Not reported |
| Pion et al. 2017 [28] | Within 6 hours | Kato-Katz | Double | Not reported |
| Clarke et al. 2018 [29] | Not reported | Quantitative polymerase chain reaction (qPCR) | Not reported | Not reported |
| Bronzan et al. 2018 [30] | Not reported | Kato-Katz | Not reported | Not reported |
| Hürlimann et al. 2018 [31] | Not reported | Kato-Katz; ether-concentration method | Double | 10% were re-examined |
| Dunn et al. 2019 [32] | Not reported | Kato-Katz | Not reported | Not reported |
| Lemos et al. 2019 [33] | Not reported | Kato-Katz | Not reported | Not reported |
| Mwandawiro et al. 2019 [34] | Within 24 hours | Kato-Katz | Double | Not reported |
| Vas Nery et al. 2019 [35] | Within 4 hours | Multiplex real-time quantitative polymerase chain reaction (qPCR) | Not reported | Not reported |
| Pullan et al. 2019 [36] | Within 24 hours | Kato-Katz | Not reported | 10% were re-read |
| Loukouri et al. 2020 [37] | Within 24 hours | Kato-Katz | Double | 10% were re-read |
| Chen et al. 2021 [38] | Not reported | The Kato-Katz | Single | Not reported |
| Pion et al. 2021 [39] | within 24 hours | Kato-Katz | Double | Not reported |
| Eneanya et al. 2021 [40] | Not reported | Kato-Katz | Double | 10% were re-read |
| Eneanya et al 2022 [41] | Not reported | Kato-Katz | Double | 10% were re-read |
| Gebrezgabiher et al. 2022 [42] | Not reported | Kato-Katz | Not reported | Not reported |
| Landeryou et al. 2022 [43] | Not reported | Kato-Katz | Double | Not reported |
| Muslim & Lim, 2022 [44] | Not reported | Kato-Katz; direct wet smear; formalin-ether concentration technique | Not reported | Not reported |
| Pion et al. 2022 [45] | Within 24 hours | Kato-Katz | Double | Not reported |
| Dyer et al. 2023 [46] | Not reported | Quantitative polymerase chain reaction (qPCR) | Not reported | Not reported |
| Le et al. 2023a [47] | Fixed for up to 10 weeks | Quantitative polymerase chain reaction (qPCR) | Not reported | Not reported |
| Le et al. 2023b [48] | Fixed for 4 days | Quantitative polymerase chain reaction (qPCR) | Not reported | Not reported |

prevalence following PC [40,41]. Authors suggest this may be caused by a higher re-infection of *A. Lumbricoides* compared to other helminths [40]. This may also be linked to the disruption in PC due to the Ebola outbreak. For *T. Trichiura*, the study from Togo reported a significant increase in prevalence following five years of annual PC (RR: 21.31, 95% CI: 15.59, 29.13)

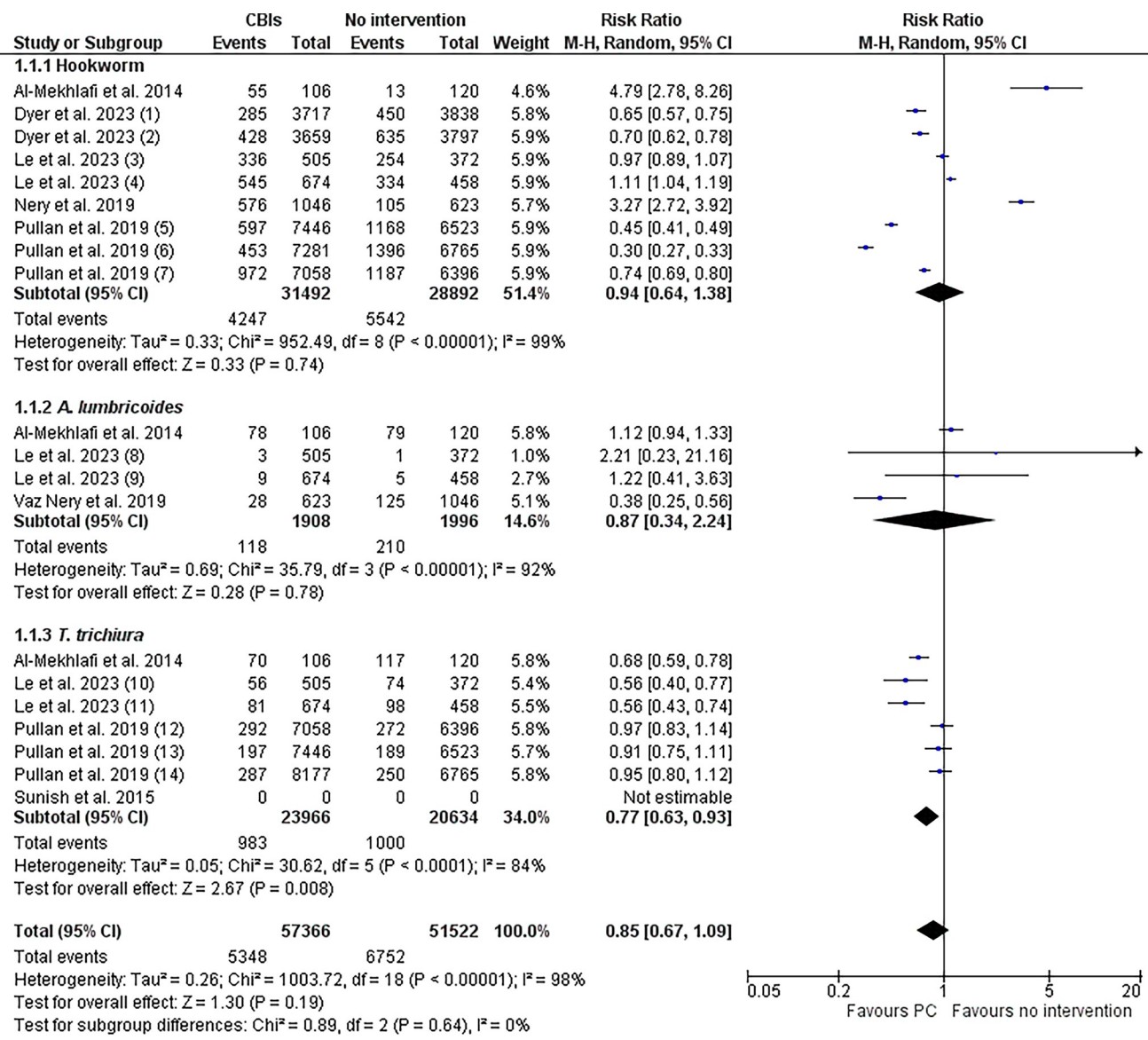

**Fig 2. Forest plot showing the impact of PC on the prevalence of STHs.**

[30]. The authors noted that *T. Trichiura* infection was very rare, likely due to a long history of onchocerciasis-focussed ivermectin distribution. 97% of STH infections were accounted for by Hookworm infections [30]. The study had serious risk of bias as the slide reading time foreseen in the protocol was not always respected, and convenience sampling was used. Paige and colleagues assessed the efficacy of a sandal (*Holoflop*) with educational imagery attached in a community receiving regular school-based deworming [27]. They found no significant reduction in STH prevalence.

## Effect of PC on intensity of infection

This meta-analysis indicates that PC is also effective in reducing infection intensity for STHs overall (SMD -387.69 95% CI: [-640.76, -134.61]), as well as for all species (Fig 3). The largest reduction in intensity was found for *A. lumbricoides* (SMD -2723.34 95% CI: [-5014.85, -431.84]), followed by *T. trichiura* (SMD -736.69 95% CI: [-1349.97, -123.42]). The forest plot shows that the CI of pooled estimate for Hookworm cross zero, suggesting no significant impact on infection intensity (Fig 3). The CI of pooled estimates for *A. lumbricoides* and *T. Trichiura* do not cross zero. The mean values of *A. lumbricoides* intensity were significantly higher than those reported for the other two species of STH, indicating higher infection rates. Most endline intensities reported were of low or medium classification.

Few studies reported differing effects of CBIs on prevalence and intensity. Eneanya et al found prevalence of *A. lumbricoides* to increase following annual [40,41] and semi-annual [40]

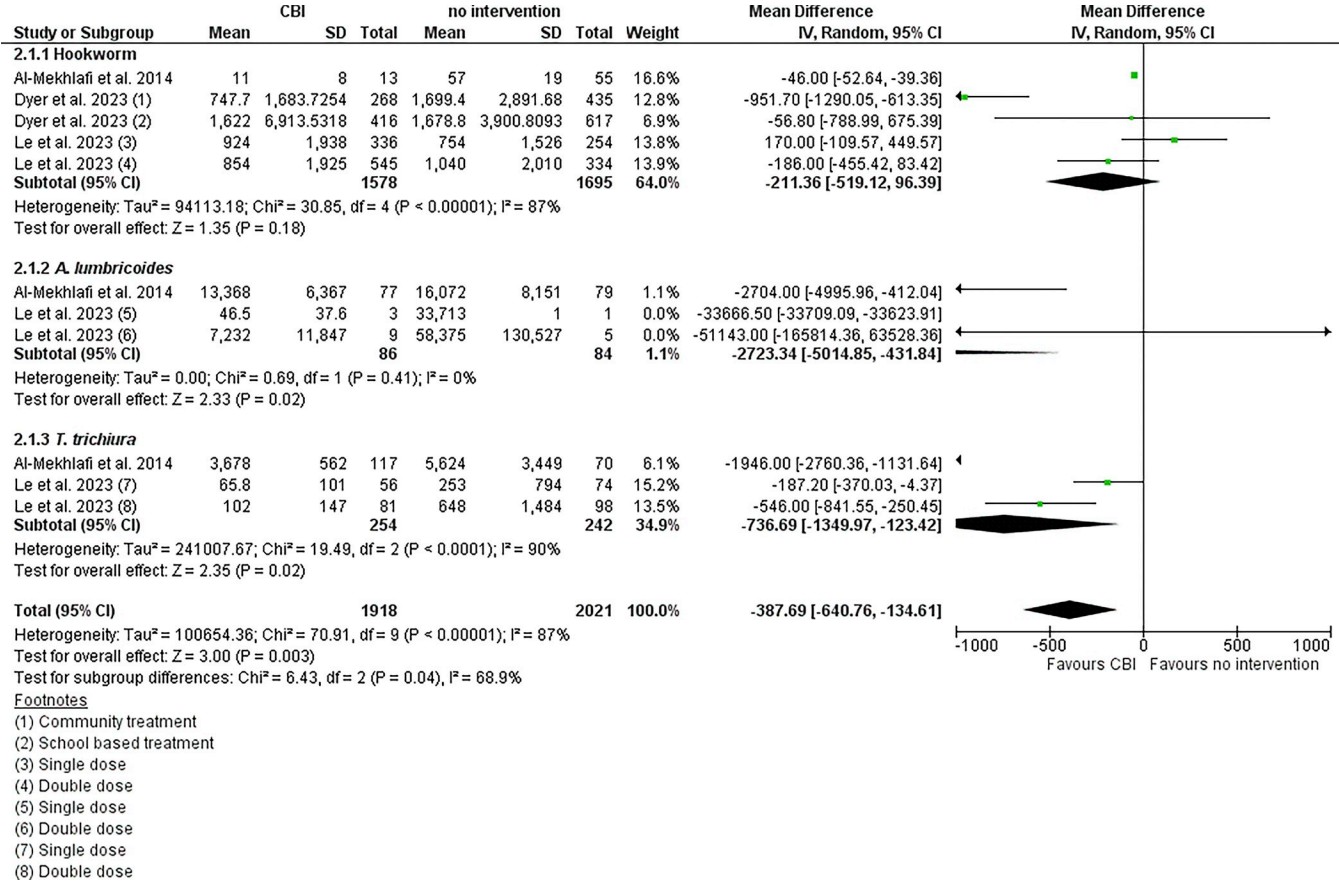

**Fig 3. Forest plot of the impact of PC on STH infection intensity.**

community-wide PC, yet the intervention was associated with decreases in infection intensities (geometric mean epg) ($P < 0.001$ in both annual and semi-annual treatment) [40]. All other studies were consistent or did not report both prevalence and intensity.

**Effect of WASH and health education on STH prevalence and intensity.** Two RCTs investigated the effect of WASH interventions on the prevalence of STHs, while only one trial investigated a health education package (which was a school-based intervention). One trial did not report the enrolment numbers per treatment arm or baseline/ endline [16]. This was a cluster-RCT in Kenya assessing the impact of school-based WASH that only found a significant effect on *A. lumbricoides* prevalence and intensity, but not on *T. trichiura* or hookworm [16]. The other trial reported no statistically significant difference on infection with *Ascaris* spp. (relative risk [RR] 2.87, 95% confidence interval [CI]: 0.66–12.48, $P = 0.159$) or *Necator americanus* (RR 0.99, 95% CI: 0.52–1.89, $P = 0.987$) with integrated WASH and deworming compared to deworming alone.

The trial investigating the effect of health education reported no statistically significant difference in prevalence of STH infection [17]. However, the study reported a significant reduction in *A. lumbricoides* infection intensity only (RR = 0·42; 95% CI = 0·21 to 0·85).

## Semi-annual versus annual mass drug administration

A subgroup analysis was conducted to compare semi-annual vs annual PC for each STH species as well as for all STH species combined (Table 3). Individual studies comparing semi-annual and annual PC all have similar STH prevalence at baseline. The number of studies included in annual subgroup analysis varied between five to eight, with a total of 20 studies evaluating semi-annual PC and 19 studies annual PC. Pooled analysis for STHs shows no statistically significant difference in prevalence reduction between semi-annual (0.38 [0.28, 0.53]) and annual treatment (0.53 [0.41, 0.69]) frequency. This also holds for individual species. Although species-specific prevalence reductions differed for treatment frequencies, this was not significant.

## Comparison of community wide treatment and school-based deworming

Seven studies delivered interventions in school settings, treating school children only, and 13 delivered community-wide treatment (Table 4). Both school-based and community-wide treatments are found to be effective in reducing STH prevalence. However, there is no statistically significant difference in outcomes between both treatment delivery methods. The biggest reduction in prevalence for both delivery platforms was found for hookworm (75% reduction for CWT and 83% reduction for SBT), followed by *A. lumbricoides* and *T. trichiura*. The subgroup analysis for CWT revealed a pooled STH relative risk of 0.38 (95% CI 0.29, 0.49) for CWT and 0.43 (95% CI 0.31, 0.61] for SBT. There were limited data to compare integrated versus non-integrated approaches.

**Table 3. Effect estimates (relative risk and 95% confidence interval) for semi-annual and annual preventive chemotherapy.**

| | Semi-annual Preventive Chemotherapy | | Annual Preventive Chemotherapy | |
|---|---|---|---|---|
| | Datasets included | Risk ratio, $I^2$ | Datasets included | Risk ratio, $I^2$ |
| Hookworm | 8 | 0.25 [0.20, 0.32], 86% | 7 | 0.16 [0.09, 0.28], 99% |
| *A. lumbricoides* | 5 | 0.38 [0.14, 1.02], 99% | 6 | 0.75 [0.59, 0.96], 98% |
| *T. trichiura* | 7 | 0.68 [0.51, 0.92], 96% | 6 | 0.53 [0.41, 0.69], 96% |
| STH combined | 20 | 0.38 [0.28, 0.53], 99% | 19 | 0.37 [0.27, 0.51], 96% |

**Table 4. Effect estimates (relative risk and 95% confidence interval) for CWT and SBT.**

| | Community Wide Treatment | | School Based Treatment | |
|---|---|---|---|---|
| | Datasets included | Risk ratio, $I^2$ | Datasets included | Risk ratio $I^2$ |
| Hookworm | 8 | 0.25 [0.21, 0.31], 90% | 7 | 0.17 [0.06, 0.48], 100% |
| *A. lumbricoides* | 5 | 0.51 [0.28, 0.91], 98% | 6 | 0.65 [0.51, 0.84], 98% |
| *T. trichiura* | 7 | 0.55 [0.39, 0.77], 97% | 6 | 0.72 [0.61, 0.84], 88% |
| STH combined | 20 | 0.38 [0.29, 0.49], 99% | 19 | 0.43 [0.31, 0.61], 100% |

Secondary outcomes of interest were only reported in three studies. One study reported a significant drop in the prevalence of anaemia from 55.6% to 14.5% [26]. The study also reported a significant reduction in the prevalence of stunting. The other two studies reported non-significant reduction in the prevalence of anaemia [33]. The integrated intervention, treating STHs, malaria and schistosomiasis, found strong association between anaemia and urinary schistosomiasis (*S. haematobium*) and no *A. lumbricoides* infections were found in severe cases [33].

There are too few data to determine the impact of health education on STH infection. One included study reported a significant difference in prevalence reduction for Hookworm among those who received community-led total sanitation and health education compared to those who received PC alone (Difference in proportions = -0.10 95% CI: -0.16, -0.04) [31].

## Discussion

This meta-analysis suggests that CBIs are effective in reducing both the intensity and prevalence of STH infection. Not all studies reported both the prevalence and intensity of infection, and even fewer reported prevalence of intensity categories (i.e., categories based on epg). These additional data help explore the effect of CBIs on STH infection, as changes in intensity categories may be overlooked if only assessing the mean intensity of infection.

The pooled risk ratio for the impact of PC on *T. trichiura* infection (RR: 0.77 95% CI: [0.63, 0.93]) was the lowest. The impact on *T. trichiura* prevalence and intensity was significant, as the confidence interval did not cross zero in the forest plot (Figs 2 & 3). In contrast, the impact on the prevalence of *A. lumbricoides* and the prevalence and infection intensity of Hookworm appeared to be non-significant. The significant impact observed on *T. trichiura* prevalence and infection intensity, despite the comparatively lower efficacy of existing PC drugs for this species compared to *A. lumbricoides* and Hookworm [49], is counterintuitive. This contradiction may arise because efficacy studies typically report cure rates following the administration of medications in controlled settings [11,49]. In contrast, the studies included in our meta-analysis report prevalence rates and infection intensity at defined follow-up periods after CBIs. These follow-up periods encompass not only the direct effects of the drugs but also factors such as reinfection rates and community-wide transmission dynamics. In community settings, interventions targeting other helminth species can impact the transmission dynamics of soil-transmitted helminths [30]. For example, CBIs may be implemented in regions with a long history of onchocerciasis-focused ivermectin distribution, which may account for lower *T. trichiura* infection rates [30].

The pooled results suggest a need to reconsider strategies for *A. lumbricoides and Hookworm*. However, it is important to note that when considering data from all studies included, CBIs had the greatest impact on Hookworm infection, indicating greater benefit of the intervention compared to that for *A. lumbricoides* and *T. trichiura*. One included study reported a longer time-to-reading of Kato-Katz slides than foreseen in the protocol, exceeding the time

threshold for detecting hookworm eggs, which are prone to degradation [50]. The study only reported a significant intervention effect for hookworm infection, not *A. lumbricoides* or *T. trichiura*. Though other authors did not report similar egg counting issues, it may have reduced Kato-Katz sensitivity for hookworm detection, especially as time-to-reading was not reported in many studies.

The difference in effect for Hookworm compared to *A. lumbricoides* and *T. trichiura* may indicate that re-infection occurs more quickly for *Ascaris and Trichuris*. The differences can also be attributed to differences in drug efficacy by dose, helminth species, individuals' age, or infection intensity. PC may be more effective for Hookworm compared to the other STHs. This is supported by evidence from a recent study showing that from 2000–2018 the relative prevalence of hookworm fell significantly from 30% to 5%, with less significant reductions for *A. lumbricoides* (17% to 9%) and *T. trichiura* (12% to 2%) [51].

Many studies included in the review delivered integrated control programmes, including simultaneous treatment of STHs, schistosomiasis and lymphatic filariasis. Integrated programs can reduce the burden of multiple diseases and reach many people at risk [52]. However, there are limited data to compare integrated versus non-integrated approaches in this review.

The dearth of evidence regarding the effectiveness of WASH interventions for STH also presents a notable gap in the current literature. While WASH strategies may be important in interrupting the transmission of STH, this review highlights the paucity of well-designed randomised controlled trials specifically investigating the impact of WASH interventions on STH infection. This gap in knowledge limits the ability to inform public health strategies aimed at interrupting STH transmission.

### Diagnostics

Most studies used the Kato-Katz method [53] for STH detection, which is recommended by the WHO. However, this diagnostic method is less sensitive than other methods, such as PCR, especially in regions with low infection intensity and prevalence. This is because only a small sample of stool is analysed per slide (41.7 mg) and eggs are not evenly dispersed in stool. Therefore, egg counts and prevalence at low infection intensities may be under-estimated [54]. More specific and sensitive diagnostic tools suitable for low-resource settings are required to enable a test-and-treat approach, and to help determine when PC can be stopped.

The number of slides analysed varied between studies. Most analysed the recommended duplicate stool samples, though some analysed only a single slide per participant [30]. In addition, the number of smears sometimes varied between baseline and follow-up measures, affecting primary outcome measures.

The results of the meta-analysis had an $I^2$ value ranging from 87–98% for STH indicating a high level of heterogeneity. A high level of heterogeneity was expected for this meta-analysis for several reasons. First, CBIs can be delivered at different intervals including monthly, bi-annually and annually. The included studies also have varying risk of bias with assessed quality ranging from very poor to good (see S1 Data). This assumption is validated by the lower $I^2$ observed during subgroup analysis. Other factors that can contribute to heterogeneity include variations in study population, study duration, research method and healthcare systems [14]. In future studies, a meta-regression could be conducted to explore which study characteristics explain variation between studies, such as follow-up time, intervention, publication date.

The findings are in agreement with those of the previous review conducted [10]. Very similar reductions in STH prevalence were found by Salam et al (reduction of 55% (RR: 0.45, 95% CI: 0.38, 0.54) versus a reduction of 53% here. Trichuriasis prevalence displayed the lowest reduction following CBI, 34% in Salam and 28% in this review. Hookworm and Ascariasis

prevalence was reduced by 60% (95% CI: 0.31, 0.52) and 68% (95% CI: 0.20, 0.51) respectively, whereas by 76% and 32% in this review. The smaller effect on *A. lumbricoides* prevalence in this review is partially due to the two studies from Liberia which contributed four datasets in total and reported low treatment compliance (39%-56%).

## Limitations

Several studies had multiple follow-up timepoints for measuring prevalence and intensity of infection, and endline data are not all from the same follow-up time. It was not possible to extract data for the same follow-up time as study designs and lengths differed. This may explain differences in prevalence and infection intensity between studies. Future reviews could conduct sub-group analyses to explore the impact of follow-up time on prevalence and intensity of infection.

The comparison between school-based and community-wide treatments within this review was limited by wide confidence intervals. These wider intervals underscore the variability and uncertainty surrounding the estimated effects observed in our analysis. The wide confidence intervals might be attributed to several factors, including the diverse study designs, and variations in follow-up periods across the included studies. Future studies with more standardized methodologies and larger sample sizes specifically designed to address this comparison would contribute significantly to refining our understanding of the relative effectiveness of these intervention approaches.

Furthermore, the included studies did not uniformly provide age-stratified data for prevalence and infection intensity measures. Despite efforts to extract detailed information, the absence of consistent reporting across different age groups limited our ability to conduct a comprehensive age stratified analysis.

The analysis calculates effect sizes and standard errors using raw data extracted from studies with different designs. This can introduce bias, as raw data from cluster trials are unadjusted for clustering, and the standard errors may be underestimated. Some studies only provided effects instead of raw data and were therefore not included in the meta-analysis. Future reviews could extract effect sizes, if provided, and make conversions.

Mean egg counts before and after intervention were used to assess the effect of CBIs. To avoid normalising the data variance and excluding very high intensity infections, the arithmetic mean of infection intensity was extracted instead of the geometric mean. However, this may not be the best way to assess intervention effects [55]. In addition to changes in arithmetic means, changes in infection intensity categories (e.g., very high to high) should be assessed using statistical tests.

## Conclusions

This review examined the impact of CBIs on STH prevalence and infection in sub-Saharan Africa. Community delivery platforms are increasingly being advocated for the prevention and control of many public health issues, and this review suggests that they can be effective in reducing prevalence and infection intensity of STHs. Most CBIs focussed on PC, which reduces the prevalence of STH and can prevent severe disease. However, it cannot prevent reinfection in the long term which is likely if environments continue to be contaminated due to open defecation or unimproved sanitation facilities. With the added concerns of drug resistance in areas with high drug pressure, it is key for research to focus on sustainable interventions that prevent reinfection.

The importance of comprehensive sanitation, hygiene and health education is not reflected in the number of published studies. These interventions play a role in preventing reinfection

and should be integrated into existing programs for maximum benefit. Therefore, further research is required to develop more effective strategies if the target of eliminating STHs by 2030 is to be achieved.

## Supporting information

**S1 Checklist. PRISMA checklist.**
(DOCX)

**S1 Data.** Table A. PRIMARY OUTCOME DATA EXTRACTED FROM INCLUDED STUDIES Data from each included study arranged into subtables. Row 1 of each subtable outlines the study citation. Row 2 of each subtable specifies the follow-up period. Row 3 of each subtable specifies the study design. Row 4 of each subtable outlines the intervention type. Subsequent rows report the prevalence at baseline and endline for Hookworm, *A. lumbricoided* and *T. trichiura* where available. Infection intensity and nutritional outcomes are also reported where available. Table B. QUALITY ASSESSMENT SUMMARY TABLE Domains D1 to D7 corresponds to the domains outlined in the Cochrane risk of bias tool (RoB2) and the risk of bias in non-randomized studies—of Interventions tool (ROBINS-I) (see references [12] & [13]). Table C. FULL SEARCH TERMS USED Column 1: Key search terms Column 2: Synonyms used in database search.
(DOCX)

**S2 Data. List of all screened articles.**
(XLSX)

**S3 Data. Detailed risk of bias assessment.**
(XLSX)

## Acknowledgments

We would like to thank the LSHTM librarians for their support in developing the search strategy and finding full-text papers.

## Author Contributions

**Conceptualization:** Sommy C. Ugwu.

**Data curation:** Sommy C. Ugwu.

**Formal analysis:** Sommy C. Ugwu.

**Investigation:** Sommy C. Ugwu, Michael O. Muoka.

**Methodology:** Sommy C. Ugwu, Michael O. Muoka, Clara MacLeod, Sarah Bick, Oliver Cumming.

**Supervision:** Laura Braun.

**Validation:** Sommy C. Ugwu.

**Visualization:** Sommy C. Ugwu, Michael O. Muoka.

**Writing – original draft:** Sommy C. Ugwu.

**Writing – review & editing:** Michael O. Muoka.

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
