## [Decision Letter · Decision Letter 0]

22 Aug 2023

PGPH-D-23-01261

The Impact of Community Based Interventions for the Prevention and Control of Soil-Transmitted Helminths in sub-Saharan Africa: A Systematic Review and Meta-analysis

Dear Dr. Ugwu,

Thank you for submitting your manuscript to PLOS Global Public Health. After careful consideration, we feel that it has merit but does not fully meet PLOS Global Public Health’s publication criteria as it currently stands. Therefore, we invite you to submit a revised version of the manuscript that addresses the points raised during the review process.

We look forward to receiving your revised manuscript.

Kind regards,

Humayun Kabir

Academic Editor

Journal Requirements:

1. Please provide separate figure files in .tif or .eps format only and remove any figures embedded in your manuscript file. Please also ensure all files are under our size limit of 10MB.

2. We have noticed that you have uploaded Supporting Information files, but you have not included a list of legends. Please add a full list of legends for your Supporting Information files after the references list. 

Additional Editor Comments (if provided):

Reviewers' comments:

Reviewer's Responses to Questions

**Comments to the Author**

1. Does this manuscript meet PLOS Global Public Health’s publication criteria? Is the manuscript technically sound, and do the data support the conclusions? The manuscript must describe methodologically and ethically rigorous research with conclusions that are appropriately drawn based on the data presented.

Reviewer #1: Yes

Reviewer #2: Yes

Reviewer #3: Partly

2. Has the statistical analysis been performed appropriately and rigorously?

Reviewer #1: Yes

Reviewer #2: Yes

Reviewer #3: No

3. Have the authors made all data underlying the findings in their manuscript fully available (please refer to the Data Availability Statement at the start of the manuscript PDF file)?

Reviewer #1: Yes

Reviewer #2: No

Reviewer #3: No

4. Is the manuscript presented in an intelligible fashion and written in standard English?

Reviewer #1: Yes

Reviewer #2: Yes

Reviewer #3: Yes

5. Review Comments to the Author

Reviewer #1: This is an interesting systematic review and meta-analysis that complies with PRISMA guidelines in reporting the results of an updated review of the impact of CBIs for STH. While new data are presented, there are a number of issues with this review that should be addressed.

It would be standard practice to exclude non randomized or pseudorandomized trials from such a review. The inclusion of other study designs introduces potential bias and is difficult to interpret. Would suggest the primary analysis should only include RCTs.

It is not at all clear why CBIs would be lumped together for such a review. There is no real justification for why one would choose to group such disparate interventions together. Given the huge heterogeneity – which makes interpretation of the meta-analysis pretty much impossible, this is a huge issue. Without adequate justification, there is not a compelling sense that these should be combined.

The outcome (prevalence and intensity of STH) is highly dependent on the sampled population. It is critical to separate studies where the population sampled is among individuals treated vs. the whole population. For example, school based delivery that only samples school going children should not be included together with school based delivery in which the whole population (including adults is sampled). Likewise, community based delivery that measures prevalence in the entire population should not be included with prevalence estimated in children.

Note that the abstract needs updating – some STH are acquired from larvae in soil. In addition to MDA, hygiene and sanitation (WASH) programs are also implemented.

Please clarify whether LF program data were included – and if so, note those studies separately.

Please note that WHO guidelines encourage STARTING MDA if prevalence exceeds 20% but once started it should be continued even when prevalence is lower.

Why did the authors limit to SSA? There are many Asian studies that could have been included. There is no solid rationale provided.

Please describe the differences in Kato-Katz methods between studies – perhaps include in table. Time to read, QA/QC, etc. This has an enormous impact on hookworm detection in particular.

Figure 1 (flowchart) should be updated with more information – for example, why were 8 not retrieved? What is RoB?

Table 1 Njenga et al 2014 – why is Quality listed as PC?

Suggest age stratified prevalence and intensity be reported.

The comparison of six monthly versus annual tx must be adjusted for baseline prevalence (or stratified by prevalence) as guidelines recommend frequency dependent on baseline prevalence.

The comparisons of community to school based delivery are limited by wide confidence intervals – this should be discussed.

More discussion regarding the available lack of evidence for the benefit of WASH interventions for STH is needed.

Reviewer #2: The manuscript is well written, but there are areas that the authors may consider strengthening before the manuscript being accepted for publication. These areas are highlighted along with my comments in the attached file.

Regarding data availability, it seems the search terms that the authors indicated having been uploaded cannot be found in the submission package. Can the authors check if it is uploaded successfully at the revision phase?

Reviewer #3: MAJOR

I don't think you can pool prevalence from different kinds of interventions (unless you did some modeling and accounted for heterogeneity). What's only common among them is that they're "community-based" and these interventions would be very different from each other. Perhaps you can pool “endline” prevalence from the similar interventions

CBI definition is strange – why combine school-based interventions with community-wide treatment?

Underlying data that allowed the calculation should be included as supplement for transparency. The data should be detailed enough to allow reproducibility

OTHERS:

The search was made more than a year ago – please update

Too many abbreviations – limit it to minimum, perhaps no more than 5 abbreviations

Include search strings either as a table or a supplement

Reasons for exclusion of records (n = 2,269) should be included. This 2269 should be broken down. Reasons for records not retrieved should be indicated. Actually, for each step in the PRISMA flow diagram, the reasons should be broken down.

Table 1 – why is species just STH, I thought we’re not combining? Include number of clusters

For all tables - All abbreviations in the table should be spelled out below the table

Please proof read – why is there an intervention “school-based” and quality as “PC”

Should be explained below the table reason for quality

This is really not just an update from Salam since yours only covered sub-Saharan Africa

6. PLOS authors have the option to publish the peer review history of their article (what does this mean?). If published, this will include your full peer review and any attached files.

**Do you want your identity to be public for this peer review?** For information about this choice, including consent withdrawal, please see our Privacy Policy.

Reviewer #1: No

Reviewer #2: **Yes: **Dehao Chen

Reviewer #3: No

---

## [Decision Letter · Decision Letter 1]

25 Jun 2024

PGPH-D-23-01261R1

The Impact of Community Based Interventions for the Prevention and Control of Soil-Transmitted Helminths: A Systematic Review and Meta-analysis

Dear Dr. Ugwu,

Thank you for submitting your manuscript to PLOS Global Public Health. After careful consideration, we feel that it has merit but does not fully meet PLOS Global Public Health’s publication criteria as it currently stands. Therefore, we invite you to submit a revised version of the manuscript that addresses the points raised during the review process.

EDITOR: Please insert comments here and delete this placeholder text when finished. Be sure to:

The authors have done a very commendable job in responding to the reviewer comments and the paper is much improved. I believe that the current manuscript will be a valuable addition to the existing literature.

A few minor concerns that remain;

1) The forest plots suggest that with the exception of trichuris, the CI of the pooled estimate of effect crosses zero, suggesting no significant impact on either prevalence or intensity. The narrative of the results and discussion should be updated to reflect this.

2) Given the above lack of significant effect, I am not clear that a subgroup analysis is warranted?

3) Available data suggest that the efficacy for existing PC drugs (albendazole and mebendazole) are LEAST effective for Trichuris - yet this is where the greatest impact appears to be seen. This is counterintuitive and deserves discussion

Please ensure that your decision is justified on PLOS Global Public Health’s publication criteria and not, for example, on novelty or perceived impact.

We look forward to receiving your revised manuscript.

Kind regards,

Khadime Sylla

Academic Editor

Journal Requirements:

Additional Editor Comments (if provided):

Minor revision required

Reviewers' comments:

Reviewer's Responses to Questions

**Comments to the Author**

1. If the authors have adequately addressed your comments raised in a previous round of review and you feel that this manuscript is now acceptable for publication, you may indicate that here to bypass the “Comments to the Author” section, enter your conflict of interest statement in the “Confidential to Editor” section, and submit your "Accept" recommendation.

Reviewer #1: All comments have been addressed

Reviewer #2: All comments have been addressed

2. Does this manuscript meet PLOS Global Public Health’s publication criteria? Is the manuscript technically sound, and do the data support the conclusions? The manuscript must describe methodologically and ethically rigorous research with conclusions that are appropriately drawn based on the data presented.

Reviewer #1: Yes

Reviewer #2: (No Response)

3. Has the statistical analysis been performed appropriately and rigorously?

Reviewer #1: Yes

Reviewer #2: (No Response)

4. Have the authors made all data underlying the findings in their manuscript fully available (please refer to the Data Availability Statement at the start of the manuscript PDF file)?

Reviewer #1: Yes

Reviewer #2: (No Response)

5. Is the manuscript presented in an intelligible fashion and written in standard English?

Reviewer #1: Yes

Reviewer #2: (No Response)

6. Review Comments to the Author

Reviewer #1: The authors have done a very commendable job in responding to the reviewer comments and the paper is much improved. I believe that the current manuscript will be a valuable addition to the existing literature.

A few minor concerns that remain;

1) The forest plots suggest that with the exception of trichuris, the CI of the pooled estimate of effect crosses zero, suggesting no significant impact on either prevalence or intensity. The narrative of the results and discussion should be updated to reflect this.

2) Given the above lack of significant effect, I am not clear that a subgroup analysis is warranted?

3) Available data suggest that the efficacy for existing PC drugs (albendazole and mebendazole) are LEAST effective for Trichuris - yet this is where the greatest impact appears to be seen. This is counterintuitive and deserves discussion.

Reviewer #2: I appreciate the authors' thorough revision based on the reviewers' comments.

Under "Declarations", it seems the quotation mark in this sentence should be removed - The authors declare that they have no competing interests" in this section.

7. PLOS authors have the option to publish the peer review history of their article (what does this mean?). If published, this will include your full peer review and any attached files.

**Do you want your identity to be public for this peer review?** For information about this choice, including consent withdrawal, please see our Privacy Policy.

Reviewer #1: **Yes: **Judd Walson, MD, MPH

Reviewer #2: No

---

## [Editor Report · Decision Letter 2]

26 Aug 2024

The Impact of Community Based Interventions for the Prevention and Control of Soil-Transmitted Helminths: A Systematic Review and Meta-analysis

PGPH-D-23-01261R2

Dear Dr Ugwu,

We are pleased to inform you that your manuscript 'The Impact of Community Based Interventions for the Prevention and Control of Soil-Transmitted Helminths: A Systematic Review and Meta-analysis' has been provisionally accepted for publication in PLOS Global Public Health.

Best regards,

Khadime Sylla

Academic Editor

Manuscript accepted.